# The Influence of SARS-CoV-2 Infection on the Thyroid Gland

**DOI:** 10.3390/biomedicines11020614

**Published:** 2023-02-18

**Authors:** Aleksandra Piekarska, Marta Góral, Marta Kozula, Aleksandra Jawiarczyk-Przybyłowska, Katarzyna Zawadzka, Marek Bolanowski

**Affiliations:** 1Department of Endocrinology, Diabetes and Isotope Therapy, Students Research Association, Wroclaw Medical University, Pasteura Bank 4, 50-367 Wroclaw, Poland; 2Department and Clinic of Endocrinology, Diabetes and Isotope Therapy, Wrocław Medical University, Pasteura Bank 4, 50-367 Wroclaw, Poland

**Keywords:** thyroid, COVID-19, SARS-CoV-2, hormones, thyroiditis, hypothyroidism, hyperthyroidism

## Abstract

It is important to acknowledge the impact that COVID-19 has on the thyroid gland and how the thyroid gland status before and during infection affects SARS-CoV-2 severity. To this day those dependencies are not fully understood. It is known that the virus uses angiotensin-converting enzyme-2 as the receptor for cellular entry and it can lead to multiple organ failures due to a cytokine storm. Levels of proinflammatory molecules (such as cytokines and chemokines) which are commonly elevated during infection were significantly higher in observed SARS-CoV-2-positive patients. In terms of hypothyroidism, hyperthyroidism, and autoimmune thyroid diseases, there is no proof that those dysfunctions have a direct impact on the more severe courses of COVID-19. Regarding hyper- and hypothyroidism there was no consequential dependency between the frequency of SARS-CoV-2 infection morbidity and more severe post-infectious complications. When it comes to autoimmune thyroid diseases, more evaluation has to be performed due to the unclear relation with the level of antibodies commonly checked in those illnesses and its binding with the mentioned before virus. Nonetheless, based on analyzed works we found that COVID-19 can trigger the immune system and cause its hyperactivity, sometimes leading to the new onset of autoimmune disorders. We also noticed more acute SARS-CoV-2 courses in patients with mainly reduced free triiodothyronine serum levels, which in the future, might be used as a mortality indicating factor regarding SARS-CoV-2-positive patients. Considering subacute thyroiditis (SAT), no statistically important data proving its direct correlation with COVID-19 infection has been found. Nevertheless, taking into account the fact that SAT is triggered by respiratory tract viral infections, it might be that SARS-CoV-2 can cause it too. There are many heterogenous figures in the symptoms, annual morbidity distribution, and frequency of new cases, so this topic requires further evaluation.

## 1. Introduction

In December 2019 the World Health Organization (WHO) was informed of a few cases of lung infections (manifesting as pneumonia) caused by an unknown (at that time) pathogen from the city of Wuhan, Hubei in China [1]. Later in March 2020, WHO announced the global pandemic situation secondary to severe acute respiratory syndrome coronavirus 2 (SARS-CoV-2) [2]. As of 27 July 2022, a total of 568,773,510 patients have been confirmed with COVID-19 infection with the number of deaths equaling 6,381,643 [3].

Human coronaviruses are part of the large virus family. Most of them are benign, apart from three—the SARS (in 2002–2003), the MERS: Middle East respiratory syndrome (in 2012), and now COVID-19 (2019–present) leading to respiratory tract infections varying in symptoms—from asymptomatic to mild, up to fatal diseases including multiple organ injury and acute respiratory distress syndrome (ARDS) [4,5,6,7,8]. It is known that COVID-19 affects not only the lungs (a major target organ) but also the kidneys, intestines, liver, pancreas, heart, and many others [9,10,11]. The capacity of virus dissemination in various organs is caused by the widespread presence of angiotensin-converting enzyme-2 receptors (ACE-2) which are entry points into human cells for the SARS-CoV-2 [4,6,12,13,14,15,16]. As these receptors are highly expressed in follicular cells (the expression according to Caron is even greater in the thyroid gland than in the lungs), there is a high possibility that the thyroid gland is a target for viruses too [8,9,17,18]. Studies claim that COVID-19 infection may also lead to Th1/Th17 hyperactivity which is a possible trigger factor for inflammatory reactions resulting in the onset of some autoimmune disorders [13].

In this review, we analyze the impact of coronavirus-2 infection on the thyroid gland considering the pathogenesis, the severity of infection based mainly on the thyroid hormone levels, the presence of subacute thyroiditis (SAT), and hypo- and hyperthyroidism, taking into account autoimmune disorders too. To date, the literature presents many publications discussing this topic, however, in our manuscript there is a detailed and most current discussion about the relationship of subacute thyroiditis and SARS-CoV-2 infection. A large part of the review is focused on the very important issue of predicting the course of COVID-19 to the correlation of fT3 concentration and also its influence on autoimmune thyroid disorders.

## 2. Study Selection

A systematic search of the literature was carried out with three different databases including PubMed, Scopus, and Web of Science. The initial search, using keywords “COVID-19 and thyroid gland” showed the following number of results: PubMed *n* = 550, Scopus *n* = 595, Web of Science *n* = 426. Among 1571 works in total, we removed articles due to duplication, works concerning cancer and vaccination, lacking useful conclusions and single case reports, editorial letters, and research letters that did not apply to our work. Finally, we assessed 76 articles which were later analyzed. The study attrition diagram is depicted in Figure 1.

## 3. Pathophysiology

SARS-CoV-2 is a pathogen mainly transmitted by small liquid particles (cough, sneeze) and the presence of those infectious particles that pass through the air. It is easily spread in crowded indoor spaces with poor ventilation, especially when people are at a conversational distance. According to the WHO, people can be also infected by touching their eyes and nose after direct contact with objects contaminated by the virus. Considering the pathogenesis of COVID-19 it is important to underline the role of the ACE-2 receptor associated with transmembrane serine protease 2 (TMPRSS2) as a molecular key point for the virus to infect human cells [19]. An additional topic that must be inspected is the role of the “cytokine storm” and individual chemokines and cytokines involved in its onset.

## 4. ACE-2 and TMPRSS2

Angiotensin converting enzyme-2 is a transmembrane glycoprotein localized on the cell surface and it is primarily involved with angiotensin ontogenesis [11]. Angiotensin itself is a peptide hormone causing an increase in blood pressure mainly by vasoconstriction. Different ACE-2 expression levels are observed in many human organs. The most noticeable expression is detected in the lungs, epithelium, small intestine enterocytes, liver, kidneys, heart, adipose tissue, prostate, testes, ovaries, and endocrine organs such as the thyroid, adrenal glands, pituitary, and hypothalamus [9,10,11,18,20,21]. According to Croce et al., ACE-2 is also present in some immune cells: macrophages, monocytes, and dendritic cells [22].

The viral S protein recognizes ACE-2 and plays a receptor-like role for cellular entry, followed by efficient binding and membrane fusion, which is associated with TMPRSS2 receptors that help S-protein priming [11,20,23]. What is worth mentioning is that D’Ardes et al. claim that the expression of ACE-2 and TMPRSS2 is higher in the thyroid gland compared to tissues in the respiratory tract [19]. Later, in the epithelium, this infection may lead to pyroptosis which is a form of programmed cell death associated with inflammation. During that process we can find a release of intracellular molecules including genetic material, pathogen-associated molecular patterns (PAMPs) and others [22].

## 5. Cytokine Storm

In some cases, during COVID-19 infection, cytokine storms can be observed. The pathomechanism of this phenomenon linked to SARS-CoV-2 is not fully understood but it can lead to multiple organ failure. The cytokine storm is characterized by an over-responsive immune and inflammatory reaction which can trigger epithelial and endothelial cell injury by inducing intensive apoptosis and vascular leakage. Due to the infection, the levels of proinflammatory cytokines such as interleukin-1 (IL-1), interleukin-6 (IL-6), interleukin-8 (IL-8), interleukin-17 (IL-17), chemokines, and others such as tumor necrosis factor-alpha (TNF-α) and interferon gamma (INF-γ) in the bloodstream are often much higher than normal elevated levels occurring during infection [11,13]. Those molecules are secreted by dendritic cells, macrophages, mast cells, endothelial cells, etc. [22]. While this condition has an impact on every human tissue it can also indirectly damage the thyroid gland.

## 6. Hypothyroidism

Thyroid hormones play an important role in the functioning of the human body. Any dysregulation may result in a deterioration of the immune response, which may lead to major clinical consequences [24]. The influence of pre-existing hypothyroidism on COVID-19 outcomes, which is a frequent comorbidity, remains unclear [25].

Pereira et al. investigated whether hypothyroidism is associated with a worse prognosis in COVID-19. A total of 526 patients (6.8%) from a group of 7762 COVID-19 patients were previously diagnosed with hypothyroidism. There was no difference regarding the most frequent symptoms (dyspnea, fever, and dry cough), their severity, and the median time of duration of those (six days) between the study and matched control groups. The analysis did not reveal any statistically significant dissimilarity considering laboratory examinations or treatments used during hospitalization. Interestingly, fewer patients from the hypothyroidism group compared to the control (36.1% vs. 42.0%) presented tachypnea at admission and required mechanical ventilation throughout the entire hospitalization (25.4% vs. 33.1%), which might be considered a better outcome indicator. Additionally, a lower mortality in patients with underlying thyroid disease was observed (22.1% vs. 27.0%). The findings of this study indicate that hypothyroidism should not be considered a risk factor for COVID-19 severity [26].

Van Gerwen et al. analyzed 3703 COVID-19 patients—251 (6.8%) were previously diagnosed with hypothyroidism. No crucial difference in the need for hospitalization between the study and control groups was found. Regarding the hospitalized patients, hypothyroidism was not linked either with more frequent requirements for mechanical ventilation or with an increased mortality. There was a correlation in the overall survival between the two studied groups identified. According to this study, hypothyroidism is not a risk factor associated with a poorer prognosis of the COVID-19 outcome. Therefore, additional examinations or consultations are not recommended [27].

Bogojevic et al. collected data from 20,366 COVID-19-positive patients, from which 1616 (7.9%) presented hypothyroidism. After adjusting for age, sex, BMI, smoking history, and comorbidities, no notable difference in COVID-19 severity, length of hospitalization, and mortality in patients with pre-existing thyroid conditions was observed. Underlying hypothyroidism was not a risk factor regarding the Intensive Care Unit (ICU) admission or mortality [28]. Another study confirmed the previous finding. Nguyen et al. demonstrated that there was no statistically significant difference in the length of hospitalization and mortality between patients with underlying hypothyroidism and the control group [29].

## 7. Hyperthyroidism

Hyperthyroidism leads to the overexpression of the ACE-2 and induces dysregulation of RAAS (renin-angiotensin-aldosterone system). It suggests that people with thyroid disease might have enhanced chances of acquiring SARS-CoV-2 [11]. In opposition, Nguyen et al. showed that there is no association between hyperthyroidism and an increased risk of COVID-19 infection. The infection rate among hyperthyroid patients who tested positive for COVID was not significantly different from the infection rate of non-hyperthyroid patients. This observation remained similar when analyzing men and women separately. Moreover, the hospitalization rate remains similar for patients with hyperthyroidism compared to those without [29].

Pizzocaro investigated the outcome of SARS-CoV-2 related thyrotoxicosis in survivors of COVID-19. The study involved 29 cases with primary thyrotoxicosis diagnosed after hospitalization for COVID-19 and then followed up after at least 30 days after discharge. At the beginning of the study, there were 58 enrolled patients, although 23 patients died and six were lost at the follow-up. Compared to the survivors, descendants’ free triiodothyronine (fT3) level was noticeably lower. On the follow-up, their median serum thyroid stimulating hormone (TSH) values increased and their free thyroxine (fT4) values decreased. There was no significant change in the fT3 level. Eight patients became euthyroid and in one case hypothyroidism developed. All subjects were tested negative for thyroglobulin antibodies (anti-TG), thyroid peroxidase antibodies (anti-TPO), and antibodies against the thyroid stimulating hormone receptor (anti-TSHR). To sum up, most cases of SARS-CoV-2-related thyrotoxicosis normalize within a few weeks after resolution [30].

Nakamura et al. performed an analysis of the thyroid function in Japanese patients with COVID-19. The study included 147 patients with confirmed COVID-19 and all of them were assessed for thyroid function (TSH, fT3, fT4) on admission. Due to the infection severity, patients were divided into three groups: 22 were categorized into mild, 41 into moderate, and 84 into the severe group. The analysis showed that most mild and moderate groups were euthyroid, but a low TSH level was frequently observed in the severe one. Furthermore, the median serum TSH level was significantly lower in the severe group than in the remaining (0.55 µU/L in the severe group vs 1.57 µU/L and 1.09 µU/L in the mild and moderate). Free thyroxine levels did not differ among the groups. The mortality rate of patients with normal and low TSH levels did not differ markedly [31].

## 8. Autoimmune Thyroid Diseases

Many viral infections have been considered as environmental factors and are reported to play a role in the pathogenesis of autoimmune-related thyroid disorders. There is a surmise that SARS-CoV-2 is disposed to trigger the immune system to hyperactivate latent reactions or lead to a new onset of autoimmune disorders [32]. The ability to trigger autoimmunity is associated with the cytokine storm [33]. Another origin of post-COVID thyroid autoimmunity is the high expression of ACE-2 receptors on the thyroid cells. This is used by viruses for cellular entry and facilitates subsequent thyroid diseases [11]. So far, four patients with post-COVID Graves’ disease were reported. Three out of four cases set an example of hyperactivation in latent autoimmune disease and one was related to triggering a new onset [9]. It is probable that there are more such cases, although unpublished.

Lui et al. collected data from prospective follow-ups of thyroid function and autoimmunity among COVID-19 survivors. They investigated the blood test results of 122 COVID-19 survivors. All of them had thyroid function tests (TFTs) conducted on admission and were reassessed after three months. A total of 102 patients had normal TFT results on the baseline. Among 20 patients with abnormal thyroid function tests (TFTs) on admission (mostly low fT3), 15 recovered and among 102 patients with initial normal TFTs, two had new-onset abnormalities that could represent different phases of thyroiditis. Among all patients recruited to the study, 104 patients had anti-thyroid antibody titers reevaluated and, in this group, an increase in the anti-TPO levels upon reassessment was observed. Among the 82 patients negative for anti-TPO at baseline, 19.5% (*n* = 16) had an interval increase in the anti-TPO concentration at the follow-up, of whom four patients became positive for anti-TPO. The factors associated with a significant increase in the anti-TPO levels included a worse baseline clinical severity, elevated CRP during hospitalization, and a higher baseline anti-TPO level. For 22 cases diagnosed with positive anti-TPO findings on admittance, only one patient became negative. Researchers inspected the change in the anti-TPO level according to interferon exposure, but there were not statistically significant differences between the group treated with interferon compared to the untreated group. An increase in the anti-TG titers upon reassessment was also observed. There was no noteworthy difference in the anti-TG titer increase between patients exposed and not exposed to interferon. All patients positive for anti-TG at the primary indication remained positive upon reappraise. There was no outstanding change in the anti-TSHR titer upon follow-up. An increase in the anti-TPO and anti-TG levels in three months post COVID-19 was demonstrated, which may suggest that COVID-19 has an impact on thyroid autoimmunity. It was distinct among the group with more severe COVID-19 and significant inflammatory responses during the acute illness. Noticed variations in the thyroid-related antibodies warrants further follow-ups for thyroid dysfunction among COVID-19 survivors [34].

In other research, Lui et al. explored thyroid dysfunction concerning the immune profile and disease status in 191 patients with COVID-19. On admission, TSH was measured in all 191 patients, fT4 in 188 of them, and fT3 in 178 patients. Anti-TPO and anti-TG were available in 188 patients and anti-TSHR in 183 patients. Focusing on autoimmune diseases, two patients had suppressed TSH levels, found together with high normal fT4 and fT3 as well as elevated anti-TSHR. One patient was also positive for anti-TPO, raising the possibility of subclinical hyperthyroidism due to Graves’ disease. Only one patient had raised TSH levels compatible with subclinical hypothyroidism and had a highly elevated anti-TPO level and mildly elevated anti-TG, also suggesting an autoimmune cause for subclinical hypothyroidism. Patients had been reevaluated after hospital discharge after one month. The patient who had Graves’ disease suspicion on admission had his thyroid function subsequently normalized. Other patients had persistent subclinical hypothyroidism and required initiation of thyroxine replacement. With the high anti-TPO concentration, this likely represented preexisting Hashimoto thyroiditis diagnosed during hospitalization for COVID-19. The rest of the patient’s thyroid function profiles were compatible with thyroiditis at various stages of evolution. However, it remains to be elucidated whether the viral load of SARS-CoV-2 and the presence of antithyroid antibodies are associated with the occurrence of thyroid dysfunction. Thyroid dysfunction in these cases was probably not mediated by autoimmunity triggered by SARS-CoV-2 infection. On the other hand, infections represent an environmental trigger of subsequent autoimmune thyroid diseases and thyroid dysfunction. It is known that in the months or years following subacute viral thyroiditis, there is a higher incidence of thyroid autoimmunity and hypothyroidism [35].

Santacroce et al. investigated the risk of SARS-CoV-2 infection among drug-naive patients with autoimmune disorders. Between May 1 and 21 May 2021, they carried out telephone interviews with 400 patients suffering from autoimmune disorders. A total of 100 of them were suffering from autoimmune thyroid disease, precisely 80 from autoimmune thyroiditis, and 20 with Graves’ disease. None of the patients required immunosuppressive therapies and all of them were regularly followed-up at the University Hospital of Pavia. In general, 33 out of 400 patients tested positive for COVID-19, of whom nine patients from the autoimmune thyroid disease group tested positive for SARS-CoV-2 and one of them required hospitalization. This figure is comparable to that reported in the general population in the same geographical area. The findings of this study indicate that patients with pre-existing autoimmune disorders appear to have a similar risk of SARS-CoV-2 infection compared to the general population [36].

## 9. Subacute Thyroiditis (SAT)

SAT also known as De Quervain’s thyroiditis, is a self-limited disorder caused by inflammation of the thyroid gland. Murugan et al. distinguished four phases lasting an average of four to six months in total. The initial phase is characterized by neck pain and hyperthyroidism, the second is usually asymptomatic euthyroidism, the next is a lasting couple of weeks of hypothyroidism, and the last one is the convalescence phase [32]. The pathogenesis of this disorder is mainly connected to respiratory tract viral infections. According to Abreu et al., thyroid injury is caused by T-cell activity. Before the COVID-19 pandemic, it was known that various viruses might lead to SAT, for example: CMV, EBV, coxsackievirus, adenovirus, enterovirus, influenza, mumps, measles, rubella, and parvovirus B19 [37,38,39,40]. The mainly underlined clinical signs of SAT were neck pain (radiating to the jaw and/or ear), fever, and systemic signs such as tachycardia, fatigue, and muscle aches [41,42,43].

In a prospective study, Bahçecioğlu et al. conducted work between March 2020 and July 2021. Based on their studies, the total number of SAT cases during the COVID-19 pandemic did not differ compared to the non-pandemic period. They divided patients into three main groups. The first one was, “CoV-SAT” (patients with SAT and COVID; *n* = 12), the second “Vac-SAT” (patients diagnosed with SAT within three months after vaccination; *n* = 6) and “non-CoV-SAT” (cases not associated with neither COVID-19 nor vaccination; *n* = 46). All patients were compared based on their symptoms, laboratory results, and treatments. The authors did not notice any differences in groups considering age, gender, frequency of fever, and other systemic signs such as weakness. However, they found disparities in the severity of neck pain, diminishment of TSH levels, weight loss, and median duration of symptoms in three researched groups. In Vac-SAT the weight loss was most significant, the TSH level was the most decreased, and the duration of treatment was the shortest. Nevertheless, neck pain was the least noticeable in non-CoV-SAT patients [23]. The ultrasonography imaging did show only typical features of SAT including bi- or unilateral gland enlargement and hypoechogenic areas with decreased vascularity in cases suspected of being triggered by COVID-19 [23,40,43,44].

Pirola et al. were comparing the morbidity of SAT presence before and during the pandemic time. They came to a similar conclusion that there was no increased annual frequency of SAT cases and both the symptoms and laboratory results did not differ [45]. Other studies also presented the symptomatology of SAT as similar in the pre-pandemic and pandemic periods [41,46]. Aemaz Ur Rehman et al. added that features seen on the ultrasound were typical for the De Quervain’s thyroiditis and did not contrast during the SARS-CoV-2 period [37].

According to Bahçecioğlu et al., SAT “shows a typical seasonal variation in its incidence with a higher prevalence in the summer” which refers to the Brancatella et al. studies. In the second mentioned work, researchers found that during the COVID-19 pandemic the annual timing pattern had changed. In 2020 the occurrence of SAT thyroiditis cases focused on the second and fourth quarters of the year. Additionally, more post-COVID-19 patients experienced hypothyroidism compared to pre-SARS-CoV-2 cases. Considering the symptoms and laboratory results (higher levels of fT4 and CRP and a lower level of TSH), they concluded that post-COVID subacute thyroiditis was more severe than those caused by other viral infections [47,48].

On the other hand, Caron et al. highlighted the significant role of IL-6 and the cytokine storm leading to SAT. According to this study, patients suffered from less intense neck pain probably due to lymphocytopenia related to SARS-CoV-2 infection which led to decreased lymph-plasmocytic infiltration of the thyroid gland [39]. Stasiak et al. also noticed more painless SAT cases among COVID-19 patients which required differential diagnosis with silent inflammation of the thyroid gland associated with increased anti-TPO levels and a normal ESR (erythrocyte sedimentation rate) [49]. Additionally, Trimboli et al. observed mostly moderate–mild forms of post-COVID-19 SAT in terms of clinical and biochemical presentation, which was also correlated with lessening the steroids dose for treating those patients compared to one recommended by the most recent guidelines [44].

## 10. Thyroid Hormonal Function and COVID-19 Severity

Infection caused by SARS-CoV-2 may result in significant disturbances regarding multiple organs, including the thyroid gland [50]. The level of thyroid hormones is widely investigated in relation to the COVID-19 disease. The impact of fT3, fT4, and TSH on the severity and mortality of COVID pneumonia remains unknown. It has been established that during the critical phase of the disease, thyroid hormones are often altered, returning to a normal state while recovering [33,51].

Many authors believe that the low fT3 level may be used as a COVID-19 severity prognostic factor. The reduction in free triiodothyronine plasma concentration is caused by defects in the secretion of the thyroid gland and a slower conversion of T4 to T3. Beltrao et al. divided a group of 245 patients into non-critically ill (181) and critically ill (64). The second group presented a lower fT3 level, which might be a COVID-19 outcome prognostic value. Interestingly, the fT3 x rT3 product correlated significantly with the predictive disease severity and mortality, presenting an 80% sensitivity and 57% specificity [12]. In the clinical study conducted by Schwarz et al., patients from the lowest fT3 group required more mechanical ventilation and longer ICU (Intensive Care Unit) hospitalization and demonstrated a more severe course of the disease, which resulted in a higher mortality rate. Those findings suggested that the fT3 level at admission is a strong severity and mortality predictor [52]. Guven et al. came to the same conclusion—low fT3 plasma concentration correlated negatively with the length of hospitalization and CRP level [53]. Data collected by Dutta et al. acknowledged the fT3 hormone as an independent risk factor for critical illness and the necessity for longer hospital treatments, regardless of the patient’s previous thyroid status [54]. Sparano and co-workers additionally discovered that decreased fT3 amounts were adversely associated with inflammatory markers such as white blood cells, neutrophils, CRP, procalcitonin, NT-proBNP, and IL-6, which increased the probability of an unfavorable outcome and death [7]. Other authors, likewise, suggest that the lower fT3 level plays an important role in the risk stratification for the severe SARS-CoV-2 disease [31,35,55,56,57,58]. The mortality rate among the group with normal fT3 levels was 3.6% (1/28), but in the group of patients with low fT3 serum the mortality rate came to 26.3% (31/118). In conclusion, low TSH and fT3 levels are related to the infection severity. Low fT3 is considered to be an indicator of high mortality. A thyroid function test on admission may be used as an indicator for the prognosis of disease severity [31].

The impact of the fT3 level on COVID-19 mortality itself also is the aim of various research. Lang et al. compared thyroid function in non-survivors and survivors in the moderate-to-critical state and suggested that the low fT3 level is associated with a higher all-cause mortality rate [59]. In a study conducted by Gao et al. the all-cause mortality was increased in patients with an fT3 plasma concentration below 3.1 pmol/L. The hormone amount was notably decreased in the severely ill group [60]. Dincer et al. also suggested that a lower fT3 level is a risk factor for a poor prognosis and increased mortality. Patients admitted to the ICU presented reduced hormone concentrations, which indicate a more severe disease course [61]. COVID-19 pneumonia survivors receiving medical treatment in the ICU demonstrated higher fT3 levels, unlike the case of deceased patients [51]. An fT3 score was gradually improving in COVID-19 survivors, while it remained constantly below the normal level in most non-survivors. This is another reason for considering fT3 as an adequate mortality predictor [46]. In addition, fT3 below the normal range was associated with a worse Sequential Organ Failure Assessment score, Lung Inflammation Prognostic Index, and radiological Total Severity Score as well as extended tissue damage and inflammation [62]. It is known that in an acute phase of any illness, the low fT3 is one of many defense mechanisms, which improves and goes back to the regular amount as the human body gradually manages to eliminate the virus [11]. Nonetheless, in some cases, the extent of the hormonal changes may be related to the disease severity and mortality [5].

Numerous studies investigated the impact of thyrotropin on COVID-19. Gong et al. divided a group of 150 patients regarding their TSH levels. Those with a lower hormone level presented a more severe disease course and a higher mortality rate. The authors suggested that a low TSH concentration is an independent risk factor for 90-day mortality in patients with normal fT3 and fT4 [63]. Baldelli et al. compared thyroid hormones in three categories (A—patients hospitalized for COVID pneumonia; B—patients with COVID pneumonia admitted to the ICU; C—control patients). The two first groups demonstrated decreased fT3 and TSH levels. What is more, in the ICU patients, hormone results were significantly lower than in group A. Those findings correlate with a progressive clinical deterioration and thyrotropin may be interpreted as a negative prognostic factor [64]. Another study aimed to determine whether the TSH plasma concentration differs regarding the cause of pneumonia. In COVID-positive patients the hormone level was more reduced than in non-COVID pneumonia [65]. D’Ardes also suggested a possible impact of SARS-CoV-2 on the thyroid gland, resulting in lowered TSH levels during hospital admission in comparison with a score from before the infection [19]. Several studies recommended thyrotropin plasma concentration as a prognostic factor for COVID-19 severity and mortality—the lower level is closely related to the length of hospitalization and all-cause in-hospital death [31,54,55,56,57,61,66,67].

Ballesteros et al. included 78 patients on ICU admission in their research. On day five of hospitalization, there were significant differences regarding thyroid hormone levels between COVID-19 survivors and non-survivors. The first group presented higher levels of fT3, fT4, and TSH. Furthermore, fT4 on the fifth day after ICU admission emerged as an important mortality risk factor [51]. Other authors also detected low fT4 as a prognostic component, but with a lower statistical value than the rest of the thyroid hormones [33,57]. Guven et al. established that although there were differences regarding the fT4 level between ICU and non-ICU patients, no connection to the length of hospitalization, inflammatory markers, as well as the disease severity, were found [53].

In contradiction with previous findings, Clausen et al., after analyzing a group of 116 patients, did not observe any correlation of TSH and fT4 levels with COVID-19 in 30- and 90-day mortality [68]. In a study conducted by Asghar et al., no connection between fT3 and fT4 blood amounts and disease severity was found [66]. Malik et al. detected a decreased fT3 and elevated TSH plasma concentration in COVID pneumonia compared to non-COVID pneumonia, but there were no significant differences regarding the severity of SARS-CoV-2 disease and the impact of the virus on the thyroid gland was observed in mild, moderate, and severe cases [69]. Other researchers presented results, where all the COVID-positive patients showed a lower fT3 level, irrespective of the disease severity [70]. What is more, Wang et al. suggested that the fT3 concentration decreases in pneumonia of various etiology, not only SARS-CoV-2 [65]. Sen et al. established that an altered thyroid function is present in about one third of COVID-19 patients, but there is no correlation with the disease severity [71].

The results presented by Okwor et al. differ significantly from all studies published previously. Here, COVID-19 patients demonstrated higher plasma concentrations of fT3 and TSH [72]. Zhang et al. indicated pre-existing thyroid disease as a COVID-19 severity risk factor, suggesting that those patients require more mechanical ventilation and more frequent utilization of antibiotics or glucocorticoids. Furthermore, previous thyroid disease caused longer hospitalization, more serious complications, and higher mortality rates [73] (Table 1).

## 11. Conclusions

The SARS-CoV-2 infection has an undeniable impact on human organs including the thyroid gland. Taking into consideration the fact that the virus enters the human tissues by using mainly ACE-2 receptors and it can trigger a cytokine storm by inducing chemokines and cytokines secretion, it is certain that COVID-19 can either directly or indirectly damage many structures in the human body. Currently, the pathogenesis is not fully understood but it is a crucial part of comprehending the present and distant complications of COVID-19 infection.

Scientific data regarding hypothyroidism and COVID-19 still requires further evaluation. There is considerable uncertainty concerning the impact of hypothyroidism on COVID-19 severity and mortality. Importantly, patients with thyroid disease shall continue with their treatment at the same dose as recommended before, to avoid hormonal dysregulation which might turn them into a more susceptible group [74].

Thyroid hormones play an important role in the functioning of the human body. Any dysregulations in the thyroid hormone levels may have consequences on the immune response and general health [24]. Although analyzing the enclosed research, hyperthyroidism does not seem to be a crucial issue in the risk of COVID infection or in the process of the disease. Results of thyroid function tests suggest that acquired thyroid dysfunctions are reversible with recovery from COVID-19. Nevertheless, it still requires further evaluation and should not be underestimated, because thyroid dysfunction should be considered a possible manifestation of COVID-19 [75].

In terms of subacute thyroiditis, recently published studies describing a possible correlation between COVID-19 and SAT differ in observed characteristics [76]. The direct impact of triggering SAT remains unclear, but it is suspected, including the fact that subacute thyroiditis is usually connected with respiratory tract virus infections. Some authors did not notice any differences in the clinical outcome, imaging, and frequency of morbidity during the pandemic period compared to the pre-pandemic time. Others found that moderate or asymptomatic patients had many cases of De Quervain’s thyroiditis caused by SARS-CoV-2 infection [44,49]. Certainly, this topic needs further research and analysis.

The immune system imbalance caused by COVID-19 might resonate in the illness severity. In addition, it may be a starting point for the new onset or exacerbation of previous thyroid disease [75]. The available literature is still insufficient to prove and explain the association between COVID-19 and autoimmune diseases [9]. Medical practitioners should be responsive to SARS-CoV-2-related thyroid dysfunction. Thyroid tests should be reevaluated in COVID-19 survivors, as the long-term consequences might develop.

The number of available reviews discussing the impact of COVID-19 infection on immune status is substantial, which highlights the importance of taking up the challenge of analyzing this problem.

Despite the multiple studies, the correlation between thyroid hormonal function and COVID-19 severity and mortality remains unclear. Some authors indicated the negative impact of low fT3 and TSH levels on the length of hospitalization and the disease outcome [31,55,56,57]. Furthermore, the fT4 laboratory test performed on the fifth day of admission may be a mortality prognostic factor [51]. On the contrary, in numerous studies, no link between fT3, fT4, or TSH and COVID-19 severity was observed [65,66,68,69,70,71]. In conclusion, the function of the thyroid gland may be impaired in some patients, but no mechanism has been established so far. For that reason, the plasma concentration of fT3, fT4, and TSH shall be monitored not only during the acute phase of the illness but also in the recuperation period [77].

## Figures and Tables

**Figure 1 biomedicines-11-00614-f001:**
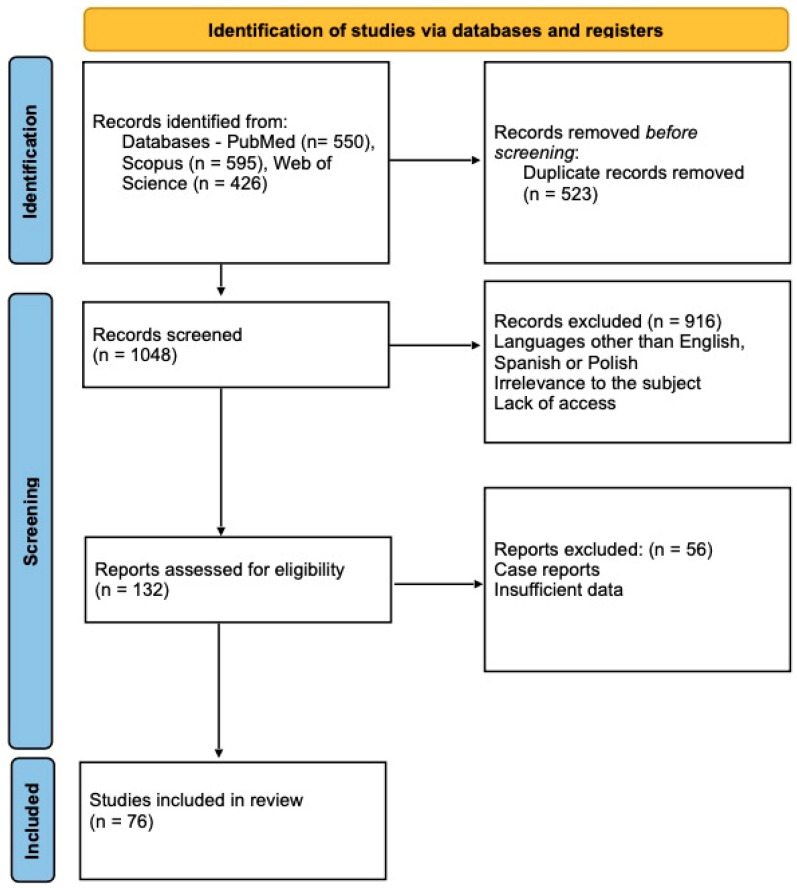
The PRISMA Flow diagram showing the study selection and attrition process.

**Table 1 biomedicines-11-00614-t001:** The characteristics of thyroid hormonal status and the course of COVID-19.

Reference Number	Author	Levels of Hormones			Hormone Levels Indicating a More Severe Course of COVID-19	COVID-19 Positive Cases	Country
		TSH	fT3	fT4			
[5]	Llamas et al.	N	↓	N	↓fT3	1183	Switzerland
[7]	Sparano et al.	N	↓	N	↓fT3	506	Italy
[12]	Beltrao et al.	N	↓	N	↓fT3	245	Brazil
[19]	D’Ardes et al.	↓	N	N	↓TSH	118	Italy
[31]	Nakamura et al.	↓	↓	N	↓fT3, TSH	147	Japan
[33]	Clarke et al.	N	↓	N	↓fT3	-	United Kingdom
[35]	Lui et al.	N	↓	N	↓fT3	191	China
[46]	Campi et al.	N	↓	N	↓fT3	144	Italy
[51]	Ballesteros Vizoso et al.	↓	↓	↓	fT4 at the fifth day after ICU admission is an important mortality risk factor	78	Spain
[52]	Schwarz et al.	N	↓	N	↓fT3	54	Israel
[53]	Guven et al.	N	↓/N	↓/N	↓fT3	250	Turkey
[54]	Dutta et al.	N	↓/N	N	↓fT3, TSH	236	India
[55]	Ayan et al.	N	N	N	↓fT3, TSH	114	Turkey
[56]	Ahn et al.	↓	↓	N	↓fT3	119	South Korea
[57]	Chen et al.	↓/N	↓/N	↓/N	↓fT3, fT4, TSH	3609	China
[58]	Zou et al.	N	N	↓	↓fT4	149	China
[59]	Lang et al.	↓	↓	N	↓fT3	127	China
[60]	Gao et al.	↓	↓	↓	↓fT3	100	China
[61]	Dincer et al.	↓	↓	↑	↓TSH	205	Turkey
[62]	Sciacchitano et al.	↓	↓	↓/N	↓fT3	62	Italy
[63]	Gong et al.	↓/N	↓	↓/N	↓TSH	150	China
[64]	Baldelli et al.	↓	↓	↓/N	↓TSH	46	Italy
[65]	Wang et al.	↓	↓	-	↓TSH	84	China
[66]	Asghar et al.	↓	N	N	↓TSH	54	Pakistan
[67]	Lania et al.	↓/N/↑	N	N/↑	↓TSH	287	Italy
[68]	Clausen et al.	↓/N/↑	-	N/↑	no correlation between mortality and TSH and fT4 levels	116	Denmark
[69]	Malik et al.	↑	↓	N	unrelated	48	Pakistan
[70]	Das et al.	↓/N	↓	↓/N/↑	unrelated	84	India
[71]	Sen et al.	↓/N/↑	↓/N/↑	↓/N/↑	unrelated	60	India
[72]	Okwor et al.	↑	↑	N	no information	45	Nigeria
[73]	Zhang et al.	↓/N/↑	↓/N/↑	↓/N/↑	pre-existing thyroid disease	71	China

↓ reduced serum levels; ↑ elevated serum levels; N serum levels within normal range.

## Data Availability

Not applicable.

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
