# Peer review of "The Influence of SARS-CoV-2 Infection on the Thyroid Gland"

_biomedicines, 2023, doi:10.3390/biomedicines11020614_

Round 1
Reviewer 1 Report
In the current manuscript, the authors have reviewed the influence of SARS-CoV-2 infection on the thyroid gland. The article is well structured into sections and subsections. English is clear and professional. It is within the scope of the journal. There are some concerns to be addressed to improve the article:
1) There are many reviews already available in the literature that are similar to the current manuscript. Authors should clearly highlight the novelty in the current manuscript. For instance, to name a few:
Scappaticcio, L., Pitoia, F., Esposito, K. et al. Impact of COVID-19 on the thyroid gland: an update. Rev Endocr Metab Disord 22, 803–815 (2021). https://doi.org/10.1007/s11154-020-09615-z (ref 78)
Inaba H, Aizawa T. Coronavirus Disease 2019 and the Thyroid - Progress and Perspectives. Front Endocrinol (Lausanne). 2021 Jun 24;12:708333. doi: 10.3389/fendo.2021.708333. PMID: 34276567; PMCID: PMC8279745. (ref 10)
Çabuk SA, Cevher AZ, Küçükardalı Y. Thyroid Function During and After COVID-19 Infection: A Review. touchREV Endocrinol. 2022 Jun;18(1):58-62. doi: 10.17925/EE.2022.18.1.58. Epub 2022 Jun 13. PMID: 35949365; PMCID: PMC9354510.
Naguib R. Potential relationships between COVID-19 and the thyroid gland: an update. J Int Med Res. 2022 Feb;50(2):3000605221082898. doi: 10.1177/03000605221082898. PMID: 35226548; PMCID: PMC8894980.
Șandru, F.; Carsote, M.; Pe tca, RC.; Gheorghisan Galateanu, AA.; Petca, A.; Valea, A.; Dumitrașcu, MC. COVID 19 related thyroid conditions (Review). Exp Ther Med 2021, 22 756. doi: 10.3892/etm.2021.10188 (ref 6)
2) Page 5, line 130-131: Rephrasing is suggested to improve clarity.
3) Page 6, line 188-201. There are many statements that are ambiguous.
Among 122 patients, 104 had increased anti-TPO levels. Similar trend was observed among groups? No groups are mentioned. Later, the authors mention 82 patients negative for anti-TPO. Authors also mention 22 cases diagnosed with positive anti-TPO on admittance.
Authors are suggested to check the details and rephrase for clarity.
4) Page 6, line 212-213: Rephrasing is suggested to improve clarity.
5) Page 9, line 353: Is it ICU patients (group B)?
6) Page 12, Reference section: Some of the references have missing information like page number or inconsistent format. Check reference 4, 20, 27, 44, 48, 50, 57, 62, 70, 72, and 73.
Reviewer 2 Report
This review is very well organised and authors have covered all the literature related to COVID-19 and pathological implication of thyroid gland. Authors are also discussing about the impact of hypothyroidism and hyperthyroidism on COVID-19 infection.
This literature review is quit novel and as per my literature knowledge it will open new avenues in the field of thyroid gland biology.
I don't have any suggestions when it comes to adding more references.
Please improve the abstract; it is not clear the conclusion of this review in the abstract.
Reviewer 3 Report
This paper is a review article of th influence of SARS-CoV-2 infection on the thyroid gland. The authors performed systemic search of the literature and selected 76 articles. This paper is a valuable review of the effects of coronavirus infection on the thyroid gland in relation to hypothyroidism, hyperthyroidism, autoimmune disease, subacute thyroiditis, and thyroid function.
In Table 1, please add the number of COVID-19 positive cases and the number of control cases, and countries in each article.
